# Identifying Key Features Associated with Excessive Fructose Intake: A Machine Learning Analysis of a Mexican Cohort

**DOI:** 10.3390/nu17223623

**Published:** 2025-11-20

**Authors:** Guadalupe Gutiérrez-Esparza, Mireya Martínez-García, María del Carmen González Salazar, Luis M. Amezcua-Guerra, Malinalli Brianza-Padilla, Tania Ramírez-delReal, Enrique Hernández-Lemus

**Affiliations:** 1“Researcher for Mexico” Program, Secretaría de Ciencia, Humanidades, Tecnología e Innovación (SECIHTI), Mexico City 03940, Mexico; tramirez@centrogeo.edu.mx; 2Diagnostic and Treatment Services, Instituto Nacional de Cardiología Ignacio Chávez, Mexico City 14080, Mexico; 3Department of Immunology, Instituto Nacional de Cardiología Ignacio Chávez, Mexico City 14080, Mexico; mireya.martinez@cardiologia.org.mx (M.M.-G.); lmamezcuag@gmail.com (L.M.A.-G.); maly.brianz@gmail.com (M.B.-P.); 4Dental Public Health Department, Division of Graduate Studies and Research, School of Dentistry, Universidad Nacional Autónoma de México, Mexico City 04510, Mexico; 5Department of Nutrition, Ambulatory Care Services, Instituto Nacional de Cardiología Ignacio Chávez, Mexico City 14080, Mexico; telesforo_13@yahoo.com.mx; 6Centro de Investigación en Ciencias de Información Geoespacial, Aguascalientes 20313, Mexico; 7Computational Genomics Division, Instituto Nacional de Medicina Genómica, Mexico City 14610, Mexico

**Keywords:** fructose, machine learning, risk factors, cluster analysis, nutritional status, dietary sugars, artificial sweeteners

## Abstract

**Background:** Excessive fructose intake has been linked to adverse metabolic outcomes, yet few studies have comprehensively described the clinical, behavioral, and nutritional patterns associated with different intake levels using machine learning. **Methods:** In this study, unsupervised and supervised algorithms were applied to a healthy Mexican cohort to examine features related to high fructose consumption, defined as intake above 25 g per day. **Results:** K-Means clustering identified three distinct profiles, with one subgroup showing less favorable anthropometric, biochemical, and behavioral characteristics. Supervised models, including Extreme Gradient Boosting, Random Forest, and Histogram-based Gradient Boosting, distinguished fructose intake levels with balanced accuracies around 80% and AUC up to 88.1%. Shapley Additive Explanations (SHAPs)-based interpretation highlighted body mass index, triglycerides, sleep duration, alcohol consumption, and anxiety indicators as features most consistently associated with high intake. **Conclusions:** These findings emphasize the multifactorial nature of fructose consumption and illustrate the utility of machine learning for uncovering dietary and metabolic patterns that warrant further investigation and may guide future nutrition-focused strategies.

## 1. Introduction

In recent decades, Mexico has experienced a marked increase in fructose consumption, primarily through sugary beverages and ultra-processed foods [1,2]. This dietary pattern has been identified as a major contributor to the rising prevalence of metabolic disorders, including obesity, type 2 diabetes, and cardiovascular disease [3]. A significant portion of daily caloric intake in the Mexican diet comes from foods and beverages rich in free sugars—particularly fructose—commonly found in industrialized drinks, snacks, and desserts [4,5]. In this context, fructose intake includes both intrinsic sources (fruits and juices) and added forms such as high-fructose corn syrup (HFCS), which is widely used as a sweetener in processed products.

Sugar-sweetened beverages represent the primary source of added sugar in the Mexican diet and account for a considerable proportion of total sugar intake [6]. The World Health Organization (WHO) recommends limiting free sugar consumption to no more than 10% of total daily energy intake, with a target of 5% for additional health benefits [7,8]. However, current sugar consumption levels in Mexico significantly exceed these recommendations. On average, added sugars account for 12.5% of total daily energy intake among the Mexican population [9].

Evidence indicates that excessive fructose intake—whether in free form or as HFCS—plays a critical role in hepatic lipogenesis and the development of metabolic dysfunction-associated steatotic liver disease, as well as other metabolic disturbances [10]. Mechanistically, fructose has been shown to suppress leptin and elevate ghrelin levels, promoting increased appetite and energy intake, which further contribute to metabolic dysregulation. These hormonal imbalances exacerbate hepatic fat accumulation and induce systemic insulin resistance, positioning high fructose consumption as a key driver in the pathogenesis of type 2 diabetes and related metabolic conditions [11].

Studies conducted in Mexico have found a direct correlation between high sugar intake and the rising incidence of metabolic diseases such as diabetes [6]. Despite increasing awareness of the metabolic consequences of excessive fructose consumption, few studies have comprehensively examined the complex interactions among diet, behavior, and metabolism using integrative analytic approaches capable of uncovering multifactorial risk patterns [12].

Recent advances in data-driven methods, particularly machine learning (ML), have expanded our ability to analyze such complexity in large-scale nutritional data. ML techniques allow for the identification of hidden patterns in population health and offer tools for the early detection of at-risk individuals, paving the way for more effective and personalized nutritional interventions. For example, Zheng et al. [13] applied Random Forest and Extra-Trees algorithms to distinguish the metabolic effects of different sugar sources, demonstrating that their impact on metabolism varies depending on their origin.

Similarly, Davies et al. [14] used a k-nearest neighbors (KNN) model to classify added sugar content in over 70,000 packaged foods with high accuracy (89%) to distinguish low, medium and high added sugar content in foods. The study highlighted the efficacy of ML in identifying sugar consumption patterns, further supporting the need for automated approaches in dietary assessments. AlYammahi et al. [15] applied artificial neural networks and non-linear regression to estimate sugar recovery from dates. Their models demonstrated high accuracy, achieving $R^{2}=0.986\pm 0.010$, thus establishing a strong correlation between operating parameters and total sugar content recovery.

Despite these advances, most previous investigations have focused on isolated relationships or relied on traditional regression-based methods, which often assume linearity and fail to capture the multidimensional complexity of dietary and metabolic data. The interplay between behavioral, biochemical, and nutritional factors underlying excessive fructose intake remains insufficiently characterized. Machine learning approaches can overcome these limitations by modeling nonlinear dependencies and identifying subtle, multivariate patterns that are not evident through conventional analyses. Recent studies have shown that ML can uncover latent dietary profiles and metabolic signatures with higher accuracy and interpretability than classical approaches, underscoring its potential for integrative nutritional research [13,14,15].

Therefore, the aim of this study was to identify clinical, behavioral, biochemical, and dietary features associated with excessive fructose consumption in a healthy Mexican adult cohort. Using machine learning–based models, we sought to distinguish individuals with high (more than 25 g per day) versus low (25 g or less) fructose intake and to examine how these features vary across subgroups with distinct metabolic and lifestyle profiles. By doing so, this research contributes to a better understanding of the multifactorial drivers of high fructose intake and offers insights that may inform targeted prevention strategies in similar populations.

## 2. Materials and Methods

### 2.1. Data

The dataset used in this study comes from the Tlalpan 2020 cohort, a research initiative carried out at the Instituto Nacional de Cardiología Ignacio Chávez (INCICH) in Mexico City, whose design and methodology are described in detail by Colin-Ramírez et al. [16]. The study was approved by the Institutional Bioethics Committee of INCICH (approval number 13-802) and follows the ethical guidelines established by the Declaration of Helsinki [17]. A total of 3156 participants were included in the dataset. Each individual provided informed consent and underwent a comprehensive evaluation that covered demographic, lifestyle, nutritional, anthropometric, biochemical, and clinical variables. Appendix A, found in the Appendix A, provides an overview of the variables included in the descriptive and analytical phases of the study.

#### 2.1.1. Clinical Measurements

Each participant underwent a physical examination in which the following anthropometric and clinical variables were recorded: weight, height, body mass index (BMI), waist circumference (WC), and systolic and diastolic blood pressure (SBP and DBP), measured following standardized protocols [18].

#### 2.1.2. Biochemical Parameters

Fasting venous blood samples were obtained from all participants following a 12-h overnight fast and processed at the Central Laboratory of the INCICH (Mexico City, Mexico), ensuring standardized quality control procedures. The biochemical analysis included key markers of glucose metabolism—such as fasting glucose, triglycerides, and the atherogenic index—as well as indicators of renal function, including serum uric acid, creatinine, and sodium levels. All assays were performed using validated clinical chemistry platforms under manufacturer-recommended protocols. Complete blood counts were performed using an LH 780 hematology analyzer (Beckman Coulter, Brea, CA, USA). All laboratory procedures were conducted in an ISO 15189-accredited laboratory, certified by the Mexican Accreditation Entity (Entidad Mexicana de Acreditación, A.C.; accreditation number CL-137; NMX-EC-15189-IMNC-2015/ISO 15189:2012), ensuring compliance with international standards for analytical quality, technical competence, traceability, and participation in external quality assessment programs.

#### 2.1.3. Behavioral Variables

In addition, participants provided details about their lifestyle habits, including smoking, alcohol consumption, physical activity, psychological stress, and sleep quality. Sleep quality was evaluated using the Spanish version of the Medical Outcomes Study Sleep Scale (MOS-Sleep), originally developed by the RAND Corporation (Santa Monica, CA, USA), which assesses sleep latency, duration, efficiency, and daytime sleepiness [19,20]. This instrument generates continuous scores ranging from 0 to 100 for each subdomain, where higher scores indicate greater levels of the construct being measured. The scale also provides a summary index which reflects overall sleep disturbances, with higher scores indicating greater impairment in sleep quality [21,22,23,24].

Psychological stress was evaluated using the validated Spanish version of the State–Trait Anxiety Inventory (STAI), originally developed by Spielberger et al. (University of South Florida, Tampa, FL, USA) and distributed by Mind Garden, Inc. (Palo Alto, CA, USA). The instrument consists of two 20-item subscales: AnxState (state anxiety) and AnxTrait (trait anxiety). The AnxState measures transient, situational anxiety, while the AnxTrait evaluates stable, long-term anxiety tendencies. Each subscale yields a continuous score ranging from 20 to 80, which was categorized into three levels: low (≤30), moderate (31–49), and high (≥50) anxiety [25]. For analytical purposes, each level was further recoded into a separate binary variable: participants classified within a specific level (low, moderate, or high) were coded as 1, while those in the remaining levels were coded as 0. This discretization allowed us to identify and model distinct stress profiles in both the unsupervised and supervised analyses.

Physical activity was assessed using the International Physical Activity Questionnaire (IPAQ), developed by the IPAQ Research Committee (international collaborative group), and categorized into three levels according to total weekly energy expenditure in metabolic equivalent minutes (MET-min/week): low (<600 MET-min/week), moderate (600–1500 MET-min/week), and high (≥1500 MET-min/week) [26]. For analytical purposes, each category was additionally recoded as a separate binary variable: participants classified as belonging to a given activity level were coded as 1, while those in the other two categories were coded as 0. This approach enabled the identification of specific behavioral profiles associated with each physical activity level.

#### 2.1.4. Dietary Intake

Dietary intake was assessed using a semiquantitative Food Frequency Questionnaire (FFQ) comprising 104 food items, developed by the National Institute of Public Health (INSP, Cuernavaca, Morelos, Mexico) [27]. Habitual intake of total energy, macronutrients (proteins, carbohydrates, total fats), fiber, simple sugars (glucose, fructose, galactose, sucrose, lactose, maltose), a detailed lipid profile (saturated, monounsaturated, polyunsaturated, animal and vegetable fats), and selected micronutrients (e.g., vitamin D) was estimated using the Nutritional Habits and Nutrient Intake Evaluation System (SNUT, version 1.0, INSP, Cuernavaca, Morelos, Mexico) [28]. Estimates were based on reported consumption frequency and standard portion sizes, using nutrient composition data from the Mexican System of Equivalent Foods (SMAE) developed by the INSP and the Mexican Ministry of Health (Mexico City, Mexico) [29]. Daily nutrient intake was calculated and expressed as grams per day (g/day). The FFQ and SMAE have been validated in Mexican populations and reflect local dietary patterns and food preparation practices [30].

For the purposes of this study, fructose intake was classified into two categories: natural fructose and added fructose. Natural fructose was defined as the monosaccharide naturally present in unprocessed foods such as fruits, vegetables, and natural fruit juices. Added fructose was defined as the monosaccharide introduced during food processing or preparation, including that found in sweetened beverages (soda or commercial juices), candies, processed cereals, and other ultra-processed foods, often in the form of free fructose, HFCS, or as part of sucrose. For foods containing sucrose, the fructose moiety was estimated considering the molecular composition of sucrose (C_12_H_22_O_11_). Since sucrose has a molecular weight of 342 g/mol and consists of one glucose unit (180 g/mol) and one fructose unit (180 g/mol) and a water molecule is lost (18 g/mol), the fructose content was calculated as: (sucrose value/342) × 180. Thus, approximately 52.6% of sucrose content was attributed to fructose.

Each food item in the FFQ was categorized accordingly, and daily intake was calculated as follows:Natural fructose intake (g/day): sum of fructose from intrinsic sources (e.g., fruits, vegetables, and natural juices), calculated according to portion size and reported frequency of consumption.Added fructose intake (g/day): sum of fructose from industrially processed foods and beverages, including free fructose, HFCS, and the fructose moiety of sucrose, based on typical composition values and reported consumption.

Total daily fructose intake was then obtained by summing both components. This operational classification enabled a more nuanced and accurate analysis of fructose consumption patterns in the study population.

To identify individuals with elevated fructose consumption, total fructose intake was categorized into two levels based on a predefined threshold. High fructose intake was defined as consumption greater than 25 g per day (>25 g/day). Participants consuming 25 g per day or less were classified as having low-to-moderate fructose intake. This binary classification was used as the primary outcome variable in the supervised machine learning models.

### 2.2. Methods and Computational Framework

The analysis followed a three-step strategy: (1) data preprocessing, (2) unsupervised clustering, and (3) supervised classification. Figure 1 illustrates the overall workflow of the study.

Phase 1: Data Preprocessing. The dataset comprises clinical, biochemical, lifestyle, and nutritional data. Furthermore, total fructose intake, along with daily consumption of natural and added fructose, was estimated using data from the SNUT. To prepare the data for analysis, continuous variables were standardized using Z-score normalization (mean = 0, standard deviation = 1), and dichotomous variables were numerically encoded. Although log-transformation of highly skewed dietary variables is a common preprocessing step, we retained the original scale after verifying that key features did not exhibit extreme skewness based on histogram inspection and skewness coefficients. Outliers (defined as data points located in sparse regions of the feature space) were identified using the Isolation Forest algorithm [31] with default parameters (n_estimators = 100, contamination = auto). This unsupervised method isolates anomalies by recursively partitioning the feature space, and observations with high anomaly scores were excluded prior to analysis. Skewness coefficients for all dietary predictors were computed (Appendix A). Most variables exhibited moderate or low skewness (|skew| < 2), whereas some fructose-source features showed marked right-skew consistent with occasional consumption patterns. This pattern reflects the expected zero-inflation of dietary data and does not compromise model performance given the use of tree-based and standardized regression models.

Phase 2: Clustering and Pattern Identification. To identify meaningful subgroups based on clinical, dietary, and biochemical similarities, K-means clustering was applied to the full set of standardized variables, without using any predefined labels related to fructose intake. The optimal number of clusters was determined using internal validation metrics, including Silhouette (which measures how well samples fit within their assigned cluster), Davies–Bouldin Index (which quantifies intra- and inter-cluster similarity), and Calinski–Harabasz Score (which evaluates the ratio of between-cluster dispersion to within-cluster dispersion). Clusters were subsequently interpreted using PCA biplots and post hoc profiling, allowing the identification of distinctive metabolic and nutritional patterns across groups.

Phase 3: Supervised Classification Models. The final phase involved supervised classification models to identify the clinical, dietary, and behavioral factors associated with fructose intake. A binary target variable was constructed based on individuals consuming more than 25 g per day were labeled as 1 (high intake), and those consuming less were labeled as 0 (low intake). Importantly, variables directly representing fructose intake were excluded from the input features in this phase to avoid information leakage. These variables were only used during unsupervised clustering and post hoc descriptive profiling.

In addition to ensemble-based models such as XGBoost, Random Forest, and HistGradientBoosting, logistic regression was implemented as a benchmark model due to its simplicity and statistical transparency. To enhance interpretability, Shapley Additive Explanations (SHAP) values were computed for the XGBoost model. SHAP quantifies the relative contribution of each feature to the model’s classification output, with positive values indicating stronger association with high fructose consumption and negative values with low consumption. Mean absolute SHAP values were used to rank features globally, while individual SHAP values illustrated local relationships between specific variables and the model’s estimated class membership.

For the supervised models, the dataset—balanced using the ADASYN algorithm—was randomly divided into training (70%) and testing (30%) subsets using stratified sampling to preserve class distribution. Hyperparameter tuning was performed through 10-fold cross-validation on the training set to ensure model stability. For XGBoost, regularization parameters (lambda, alpha) and tree-related hyperparameters (max_depth, learning_rate, subsample) were optimized via grid search. Random Forest and HistGradientBoosting models were fitted using bootstrapped samples with replacement, and model configurations were selected based on average performance metrics across folds to evaluate consistency rather than predictive superiority. Feature importance was estimated using impurity-based measures, while SHAP values were calculated for the XGBoost model to provide a class-specific, model-agnostic interpretation of feature associations. Additionally, logistic regression was included as a benchmark due to its transparency and interpretability. This model was fitted on standardized features, with class balance addressed using ADASYN prior to fitting, and penalty and regularization parameters optimized through grid search.

Model performance was evaluated on the test set using balanced accuracy, sensitivity, specificity, F1-score, and the area under the receiver operating characteristic curve (AUC). All computational analyses were performed in Google Colab (Alphabet Inc., Mountain View, CA, USA) using Python 3.12.12 (Python Software Foundation, Wilmington, DE, USA) within Jupyter Notebook 7.4.6. Data preprocessing, clustering, and supervised learning models (Random Forest, Logistic Regression, and Histogram-based Gradient Boosting) were implemented using the scikit-learn library (version 1.6.1), while xgboost (version 3.1.1) was employed for gradient boosting optimization. Data management and numerical computations relied on pandas (version 2.2.2) and numpy (version 1.26.4), respectively. SHAP values for model interpretability were obtained using the shap package (version 0.45.0). For detailed algorithmic formulations and mathematical definitions, please refer to the Appendix A.

## 3. Results

A total of 3156 participants were initially enrolled in the Tlalpan 2020 cohort, of whom 1151 with complete data and validated food frequency questionnaires were included in this analysis. At baseline, participants had a median age of 40 years (IQR: 31–46), with similar age distributions between sexes. Overall, the cohort showed a high median BMI (29 kg/$m^{2}$), with men presenting slightly higher BMI, waist circumference, and blood pressure compared to women. Low physical activity was reported by more than one-third of participants, particularly men, while women exhibited a higher prevalence of trait anxiety. Alcohol consumption was frequent (74%) and about one-third of participants were current smokers. Detailed baseline characteristics are presented in Table 1.

Biochemical assessments revealed similar glucose levels across sexes, while men had higher uric acid, creatinine, and triglyceride concentrations. Women showed slightly higher total cholesterol. Median total fructose intake was 53 g/day, with women consuming more fructose from fruits and men obtaining more from soda and artificial sources. Overall, carbohydrate intake was high, with men reporting greater consumption of carbohydrates, proteins, and fats, whereas fiber intake was comparable between sexes. Detailed values are provided in Table 1.

To address class imbalance, the dataset was balanced using ADASYN, resulting in an analytic sample of 1721 individuals. Subsequent analyses identified three distinct clusters with significant differences in metabolic profiles and fructose intake. Supervised models were then applied to examine and distinguish patterns associated with high fructose consumption, identifying key clinical, nutritional, and behavioral features as detailed in the following sections.

### 3.1. Cluster Analysis

After preprocessing, unsupervised k-means clustering was performed on standardized variables without predefined class labels, revealing three data-driven clusters (k=3). Internal validation metrics (Silhouette Score = 0.209, Davies–Bouldin Index = 1.506, Calinski–Harabasz Score = 406.8) indicated moderate separation; however, the three-cluster solution was retained for its clinical interpretability, representing low, intermediate, and high intake patterns. Cluster sizes were C0 = 484, C1 = 531, and C2 = 136 participants. Although PCA was not used for clustering, a two-dimensional projection of the first two components (explaining 65.1% of total variance; Figure 2) was used for visualization. Clustering was conducted on the complete standardized dataset to preserve the full variance structure.

Figure 2 and Figure 3 illustrate three data-driven dietary patterns identified through clustering. Cluster 0 (blue) displayed intermediate nutrient intakes and anthropometric indicators. Cluster 1 (orange) showed moderate-to-high macronutrient and energy intake with comparable anthropometric measures. Cluster 2 (red) exhibited the highest overall energy and nutrient intake, larger body size, and slightly less favorable renal biomarkers, along with a relatively greater contribution from added sugars.

To further describe the clusters, their associations with variables not included in the clustering process were examined (Table 2). Energy intake varied across clusters (*p* < 0.001): Cluster 2 had the highest median intake (≈3540 kcal/day) with proportionally greater contributions from protein, carbohydrates, and fat; Cluster 1 showed intermediate intake (≈2400 kcal/day); and Cluster 0 the lowest (≈1670 kcal/day). Fructose intake displayed a similar gradient, with Cluster 2 showing the highest and Cluster 0 the lowest levels. When adjusted per 1000 kcal, neither total, natural, nor added fructose differed significantly between clusters (both p≥0.22), suggesting that higher fructose intake in Cluster 2 reflects greater total energy intake rather than higher relative sugar contribution.

In absolute terms (g/day), intakes of natural juice, fruits, cereals, and soda were significantly higher in Cluster 2 (all p≤0.006). After adjustment for total energy intake, Cluster 2 showed the highest density of natural juice (p=0.039), whereas Cluster 0 exhibited the highest densities of added sugar sources, particularly soda, candies, and cereals per 1000 kcal (all p≤0.002). Fruit density did not differ significantly across clusters (p=0.183). Overall, Cluster 2 was characterized by a higher total energy and food volume pattern, while Cluster 0 displayed a relatively greater contribution of added sugars within a lower caloric intake.

Anthropometric variables showed a gradient consistent with the dietary patterns: body weight, waist circumference, and BMI were highest in Cluster 2, intermediate in Cluster 1, and lowest in Cluster 0 (all p≤0.042). Similarly, uric acid and serum creatinine levels were higher in Cluster 2 (*p* = 0.003 and <0.001), while glucose, total cholesterol, triglycerides, sodium, and the atherogenic index did not differ significantly between clusters (all p≥0.16). Lifestyle variables also varied, with snoring and ever-smoking being more frequent in Cluster 2 (*p* = 0.009 and 0.001), a non-significant tendency observed for current smoking (*p* = 0.079), and comparable alcohol and energy drink consumption across clusters.

The Sankey diagrams (Figure 4) provided a complementary visualization of fructose intake patterns across clusters. Each link represents the cluster-wise arithmetic mean intake of each fructose source (g/day) and its energy-adjusted equivalent (g/1000 kcal), calculated as individual intake divided by total energy (kcal/1000). Cluster 2 showed the highest absolute intake, mainly from sodas, fruits, and cereals. Cluster 1 presented intermediate contributions, whereas Cluster 0, despite lower total intake, exhibited a relatively higher proportion of added fructose after energy adjustment, indicating greater dependence on sweetened products relative to caloric intake. This graphical representation highlights both the magnitude and composition of fructose sources, complementing the numerical findings in Table 2.

Overall, Cluster 2 was characterized by higher total energy intake, larger body size, and modestly elevated renal biomarkers. Cluster 0 presented lower energy intake but a higher relative contribution from added-sugar sources, while Cluster 1 showed intermediate patterns, combining moderate intake levels with intermediate anthropometric and metabolic measures.

Unsupervised clustering revealed distinct dietary and metabolic profiles within the cohort but did not quantify the relative influence of individual variables on fructose intake. To complement this, supervised learning models were applied to identify and rank features associated with high versus low intake. This integrative strategy combined pattern discovery with feature-level interpretability, enhancing the overall understanding of fructose-related associations.

### 3.2. Supervised Algorithms and Model Interpretation

Based on the cluster-derived profiles, supervised models were applied to classify individuals with high or low fructose intake. To enhance interpretability, SHAP values from the best-performing XGBoost model were used (Figure 5). SHAP is a game-theoretic framework that decomposes model outputs into additive effects of individual features, enabling assessment of their overall relevance across the population (global interpretation) and their role in specific observations (local interpretation).

Globally, the most relevant features included BMI, fructose from natural juices, alcohol consumption, triglycerides, creatinine, smoking status, and age, encompassing clinical, dietary, and behavioral dimensions. In the high-intake group, higher BMI, fructose from natural juices and sodas, candies, triglycerides, creatinine, and smoking were most strongly associated with classification as high consumers. Conversely, negative SHAP values in the low-intake group indicated lower consumption of natural juices, triglycerides, alcohol, and smoking. Age also contributed to model differentiation, with older participants more frequently classified in the low-intake group, possibly reflecting differences in dietary patterns or health awareness.

Some variables, such as BMI, juice intake, and smoking, showed relevance in both high- and low-intake groups, reflecting the nonlinear, context-dependent structure of the XGBoost model. For example, higher BMI or greater juice consumption were associated with classification into the high-intake group, whereas moderate values aligned with lower intake. Similarly, smoking was not consistently linked to higher intake but, when combined with other factors, contributed to model differentiation toward the high-intake category.

Overall, Figure 5 identifies the most relevant features linked to fructose intake levels and illustrates both the direction and magnitude of their influence on model classification. This visualization provides a detailed view of how multiple factors jointly shape the model’s output, underscoring the importance of evaluating the overall profile of each participant rather than isolated variables.

In this analysis, the XGBoost classifier was trained using optimized hyperparameters (learning rate = 0.1, max depth = 3, n estimators = 100). The model achieved a balanced accuracy of 80.96%, an AUC of 88.08%, a sensitivity of 89.47%, a specificity of 72.45%, and an F1-score of 85.30%. These results indicate robust model performance. To complement the SHAP-based interpretation from XGBoost, Random Forest and HistGradientBoosting models were also trained. Feature importance values were extracted and normalized to allow direct comparison across models (Table 3).

Both ensemble models consistently identified anthropometric indicators—particularly BMI—as the most relevant features, with waist circumference and weight also ranking highly in HistGradientBoosting. Lifestyle and behavioral factors, including alcohol intake, snoring, and sleep quality, were additionally relevant, whereas anxiety-related variables appeared less prominent, except for high state anxiety in Random Forest. In terms of performance, both models reached balanced accuracies near 87–88% and AUC values around 87%, reflecting stable model performance under optimal parameter settings (Table 3).

SHAP analysis provided both global and individual-level interpretability for the XGBoost model, while Random Forest and HistGradientBoosting offered complementary perspectives on the relative importance of features at the population level. The SHAP summary plot (Figure 5) illustrates average associations by class, and instance-level values were examined to explore feature contributions in individual participants.

Although logistic regression performed slightly below the ensemble methods (Table 4), it provided transparent and easily interpretable coefficients. Detailed coefficients and 95% confidence intervals are presented in Table 5. The model identified associations with key dietary components (natural and added fructose sources) and lifestyle factors (alcohol intake and smoking), although only six of the fifteen top-ranked variables reached statistical significance (*p* < 0.05). These findings underscore the complementary value of integrating traditional regression with machine learning to more clearly characterize the behavioral and nutritional markers associated with excessive fructose intake.

## 4. Discussion

In the context of increasing data availability and the growing complexity of metabolic disorders, ML techniques have become valuable tools in nutritional epidemiology. Unlike traditional statistical approaches that often rely on linear assumptions or predefined hypotheses, ML enables the exploration of complex, non-linear associations among clinical, dietary, and behavioral variables [32,33,34].

In this study, ensemble-based algorithms, particularly XGBoost and Random Forest, helped uncover markers statistically associated with excessive fructose intake, spanning both metabolic indicators and behavioral factors [35,36,37]. These results highlight the complementary role of ML in enhancing conventional analyses by capturing multifactorial patterns that may remain undetected with traditional methods. The clustering analysis revealed three distinct metabolic–nutritional profiles, indicating that fructose consumption is embedded within heterogeneous dietary and metabolic contexts rather than reflecting a uniform risk pattern.

Cluster 2 was characterized by higher overall energy intake (≈3540 kcal/day) and proportionally greater consumption of protein, carbohydrates, fat, and fructose. Participants in this cluster also exhibited higher body weight, waist circumference, uric acid, and creatinine levels, together with a greater prevalence of smoking and snoring. These findings suggest that elevated fructose intake frequently co-occurs with broader patterns of high energy consumption and lifestyle behaviors linked to metabolic strain. Previous research has reported similar associations, noting that increased fructose exposure tends to parallel uric acid accumulation and early metabolic alterations [38].

Cluster 1 presented intermediate levels of energy and nutrient intake (≈2400 kcal/day), along with moderate anthropometric and biochemical indicators. Compared with Cluster 0, this group showed higher protein intake, a pattern that has been previously associated with favorable metabolic outcomes such as reduced lipogenesis and improved postprandial glucose regulation [39,40,41,42]. This dietary composition may help contextualize the relatively balanced metabolic profile observed in this cluster despite higher overall energy consumption.

Cluster 0 represented the lowest intake profile (≈1670 kcal/day), characterized by lower body weight, waist circumference, uric acid, and creatinine levels. However, when adjusted for energy, this group exhibited the highest relative proportion of added fructose, particularly from soda, candies, and cereals. This indicates that within a lower-calorie dietary pattern, added sugars contribute a larger share of total energy intake. Although the current metabolic indicators appear favorable, such dietary composition may warrant attention given its greater dependence on refined sugar sources.

These unsupervised findings provided the basis for the subsequent analytical phase, in which supervised learning techniques were used to characterize the factors most strongly associated with varying fructose consumption profiles. While clustering uncovered latent phenotypes linked to metabolic diversity, the supervised models identified the variables contributing most to distinguishing individuals with higher versus lower fructose intake. Algorithms such as XGBoost, Random Forest, and HistGradientBoosting demonstrated stable performance and consistently indicated that anthropometric measures, sleep duration, alcohol consumption, and state anxiety were among the most relevant correlates [43,44]. Collectively, these results underscore the utility of ML methods for describing complex, multidimensional relationships within dietary and behavioral data.

Our findings align with recent applications of ML in nutrition and food sciences. For instance, Davies et al. [14] applied k-nearest neighbors to estimate added sugar in packaged foods, achieving high predictive accuracy, while Kim et al. [45] used ensemble neural networks with Vis/NIR spectroscopy to estimate sugar content in citrus fruits, reporting similarly strong performance

The growing application of ML in nutrition research reflects increasing interest in the metabolic and functional implications of fructose consumption. Beyond dietary behavior, Jeong et al. [46] applied multiple algorithms, including decision trees, support vector machines, and neural networks, to classify sweeteners such as fructose, allulose, and kestose in food matrices using hyperspectral imaging, with results indicating robust model performance.

The central finding from our analysis is the differential associations of added versus natural fructose with biochemical and anthropometric indicators. Unlike traditional regression approaches that often aggregate these sources under a generalized *sugar intake* construct, our ML approach separated them and revealed that added fructose was more strongly associated with uric acid, triglycerides, and waist circumference, even after accounting for caloric intake, physical activity, and BMI. These observations are consistent with mechanistic hypotheses suggesting that hepatic de novo lipogenesis and purine metabolism may be disproportionately affected by added fructose, particularly in populations with high baseline metabolic risk [47,48,49].

Although the ensemble models outperformed logistic regression in classification, the latter offered greater transparency and interpretability. The logistic model confirmed associations with key dietary features, such as natural fructose intake, fruit, and industrialized sources (soda and candies), along with lifestyle factors including alcohol consumption and smoking. Notably, only 6 of the top 15 markers reached statistical significance (*p* < 0.05), indicating that while some variables show clearer associations, others may contribute in combination within the multivariate structure of more complex models. This underscores the complementary value of integrating traditional regression with ML techniques to interpret and refine observed patterns.

From a systems science perspective, the use of SHAP values provided a transparent view of feature importance, enhancing interpretability while preserving model performance. This approach highlights the relative associations of variables, offering insights that may support future applications in public health. In particular, the concept of consumption thresholds [50]—beyond which added fructose is hypothesized to increase metabolic risk—suggests the need for further research to evaluate their applicability in specific populations. Such evidence, once consolidated, could contribute to refining dietary guidelines and informing behavioral or policy interventions.

Our findings connect with the broader discourse on the social determinants of health. In urban contexts such as Mexico City, where processed food consumption is widespread and dietary choices are shaped by socioeconomic factors, identifying the metabolic risks associated with added sugars is a matter of public health relevance. In this sense, ML serves not only as a technical methodology but also as a tool to explore social and nutritional diagnostics.

An important avenue for future research is the design of dietary interventions tailored to the phenotypes identified in our cluster analysis. For example, Cluster 1 combined higher BMI and uric acid levels with moderate nutrient intake, suggesting a pattern of metabolic vulnerability. In this context, low-carbohydrate approaches (e.g., ketogenic diets), which emphasize higher protein and fat while limiting sugars, warrant further evaluation. Although beyond the scope of the present study, such strategies could be empirically tested in future work to determine their relevance for metabolically at-risk subgroups.

Importantly, while ML methods offer considerable advantages, their use requires epistemological caution. Data-driven insights should complement, rather than replace, causal inference and theoretical perspectives, and must be interpreted within an interdisciplinary framework that integrates computational, biological, and social dimensions.

This study illustrates the potential of ML to connect individual-level metabolic patterns with broader public health considerations. By combining algorithmic tools with established scientific knowledge and a population health perspective, future research may enhance the monitoring and understanding of dietary transitions, particularly in vulnerable groups. Such approaches could ultimately contribute to more adaptive and equitable strategies for health promotion.

## 5. Strengths and Limitations

This study has several important strengths. First, we employed both unsupervised and supervised machine learning techniques to analyze clinical, behavioral, biochemical, and nutritional markers associated with excessive fructose intake. The combination of clustering and supervised modeling provided complementary perspectives, enabling the identification of meaningful metabolic profiles and distinctions between individuals with higher and lower fructose consumption. Second, the use of SHAP values enhanced interpretability, clarifying how specific variables—such as BMI, triglycerides, alcohol intake, and sleep patterns—were linked to the model’s outputs. Third, the dataset is drawn from a well-characterized urban Mexican cohort, with detailed dietary data and validated instruments, offering a rich multidimensional view of metabolic health.

However, several limitations must be acknowledged. The cross-sectional design precludes causal inference. Key variables such as dietary intake and sleep quality were self-reported and therefore subject to recall or reporting bias. Although the ML models showed good performance, their applicability depends on data quality and context and may not directly extend to other populations. Nonetheless, even with cross-validation and regularization applied to mitigate overfitting, the complexity of ensemble models may still entail a residual overfitting risk, particularly given the moderate sample size of the cohort. Overfitting commonly arises when models capture noise or spurious correlations in the training data rather than generalizable patterns, especially when data are limited or heterogeneous [51,52].

The clustering approach was exploratory and data-driven: while it identified subgroups with distinct clinical and nutritional profiles, these clusters lack predefined clinical meaning and require validation. Accordingly, the findings should be viewed as hypothesis-generating rather than definitive. Overall, integrating advanced modeling with clinical and nutritional variables provides useful insights into patterns of excessive fructose intake and lays the groundwork for future confirmatory research.

## 6. Conclusions

While exploratory, this work provides a foundation for future research aimed at refining dietary recommendations and developing targeted interventions to reduce added sugar consumption, particularly in middle-income urban populations facing rapid dietary transitions. Beyond methodological innovation, our findings underscore the nutritional relevance of distinguishing between natural and added fructose, as their associations with metabolic and anthropometric markers were not equivalent. The results suggest that excessive fructose intake is embedded in broader dietary and behavioral profiles, reinforcing the need for integrative nutritional strategies to mitigate related health risks in vulnerable populations.

## Figures and Tables

**Figure 1 nutrients-17-03623-f001:**
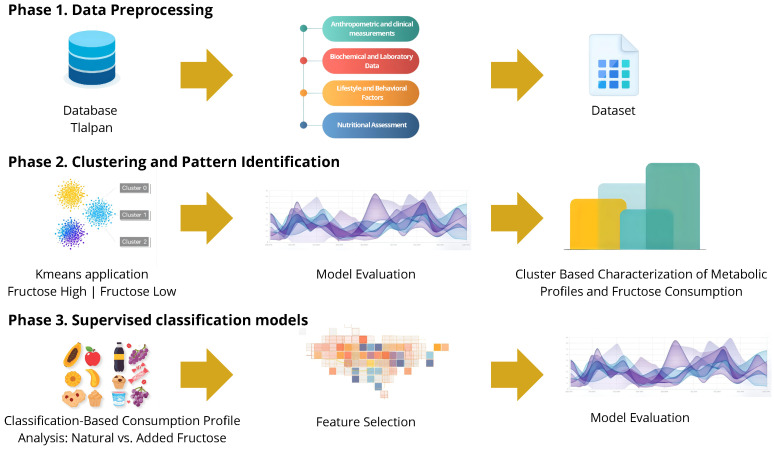
Workflow for Data Preprocessing, Clustering, and Statistical Modeling of Fructose Consumption Patterns.

**Figure 2 nutrients-17-03623-f002:**
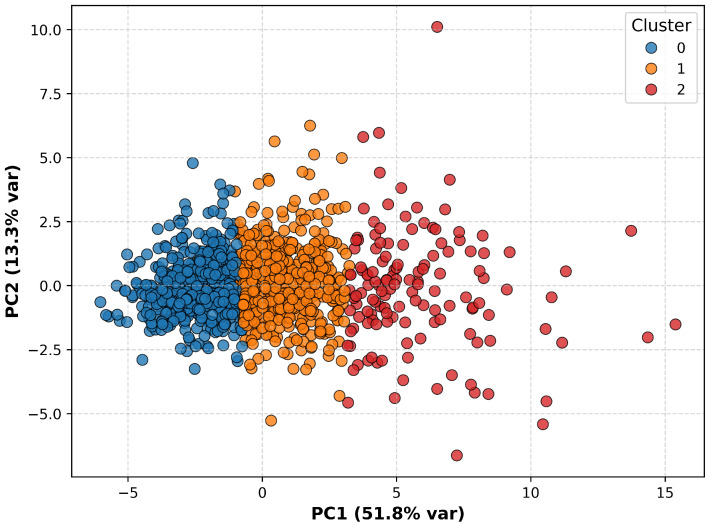
Two-dimensional principal component analysis (PCA) of standardized nutrient profiles, colored by cluster assignment. The first (PC1) and second (PC2) components explained 51.1% and 12.1% of the variance, respectively, accounting for 63.2% of the total variance.

**Figure 3 nutrients-17-03623-f003:**
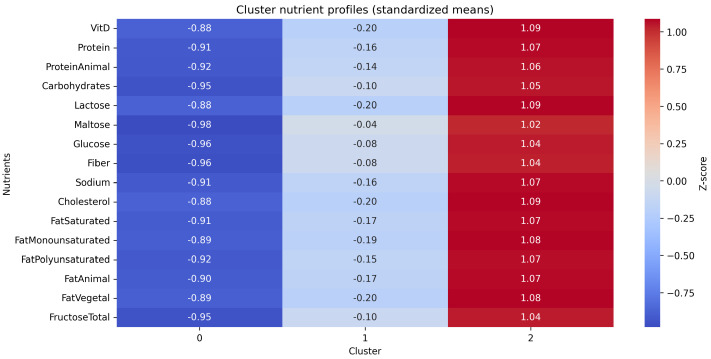
Heatmap of standardized mean nutrient values across clusters. This visualization illustrates relative differences in nutrient intake levels among clusters.VitD, vitamin D (IU); Protein, total protein (g/day); ProteinAnimal, animal protein (g/day); Carbohydrates, total carbohydrate (g/day); Lactose, lactose (g/day); Maltose, maltose (g/day); Glucose, glucose (g/day); Fiber, total fiber (g/day); Sodium, sodium (mg/day); Cholesterol, cholesterol (g/day); FatSaturated, saturated fat (g/day); FatMonounsaturated, monounsaturated fat (g/day); FatPolyunsaturated, polyunsaturated fat (g/day); FatAnimal, animal fat (g/day); FatVegetal, vegetable fat (g/day); FructoseTotal, total fructose (g/day).

**Figure 4 nutrients-17-03623-f004:**
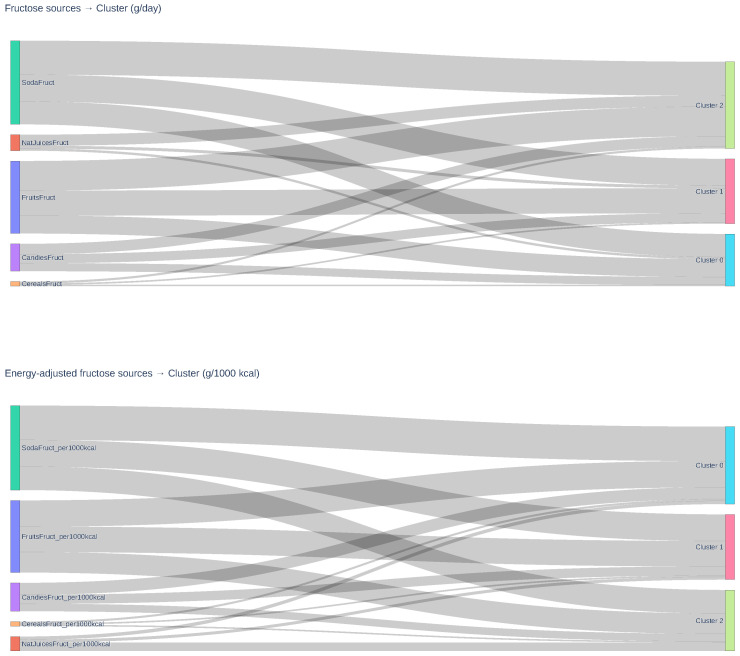
Sankey diagrams showing the contribution of fructose sources across the three identified clusters. Top panel: absolute intake (g/day). Bottom panel: energy-adjusted intake (g/1000 kcal). Cluster 1 exhibited the highest absolute intake dominated by sodas and fruits; Cluster 0 showed intermediate, more balanced contributions; and Cluster 2, despite lower total intake, displayed a higher relative contribution from added fructose. FructNat, natural fructose (g/day); FructAdded, added fructose (g/day).

**Figure 5 nutrients-17-03623-f005:**
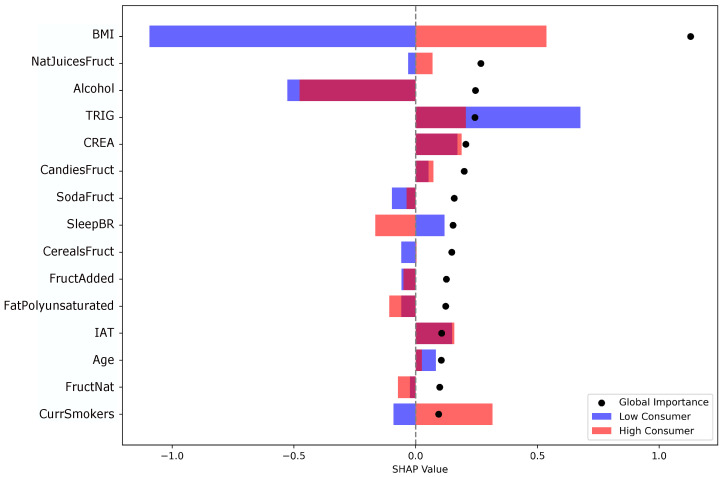
SHAP summary plot of the 15 most relevant features associated with high and low fructose intake. Bars represent the mean SHAP values for each feature by class (red: high-intake; blue: low-intake participants). Positive SHAP values indicate a stronger influence toward classification as a high-intake case, while black dots represent the overall magnitude of global importance across both classes. BMI, body mass index (kg/$m^{2}$); NatJuicesFruct, fructose from natural juices (g/day); Alcohol, alcohol consumption; TRIG, triglycerides (mg/dL); CREA, creatinine (mg/dL); CandiesFruct, fructose from candies (g/day); SodaFruct, fructose from soda (g/day); SleepBR, sleep breathing risk (0–100); CerealsFruct, fructose from cereals (g/day); FructAdded, added fructose (g/day); FatPolyunsaturated, polyunsaturated fat (g/day); IAT, atherogenic index; Age, age (years); FructNat, natural fructose (g/day); CurrSmoker, current smoker.

**Table 1 nutrients-17-03623-t001:** Baseline clinical, biochemical, behavioral, and nutritional characteristics of the study cohort, stratified by sex.

Variable	Total (n = 1151)	Women (n = 713)	Men (n = 438)
**Clinical variables** ^*a*^			
Age (years)	40 (31–46)	40 (32–46)	39 (30–45)
BMI (kg/$m^{2}$)	29.0 (26.7–31.7)	28.8 (26.4–31.7)	29.3 (27.1–31.8)
WC (cm)	95 (88–102)	92.5 (86–99)	98.5 (92–106)
SBP (mmHg)	109 (102–117)	107 (99–115)	113 (107–121)
DBP (mmHg)	73 (68–80)	72 (66–79)	78 (71–83)
**Behavioral variables** ^*b*^			
Physical activity (low), n (%)	415 (36.1)	236 (33.1)	179 (40.9)
Physical activity (moderate), n (%)	379 (32.9)	260 (36.5)	119 (27.2)
High anxiety (state), n (%)	366 (31.8)	236 (33.1)	130 (29.7)
High anxiety (trait), n (%)	377 (32.8)	278 (39.0)	99 (22.6)
Current smoker, n (%)	376 (32.7)	228 (32.0)	148 (33.8)
Alcohol consumption, n (%)	851 (73.9)	512 (71.8)	339 (77.4)
**Biochemical variables** ^*a*^			
Glucose (mg/dL)	94 (89–101)	94 (88–101)	95 (90–101)
Uric acid (mg/dL)	5.4 (4.6–6.5)	4.9 (4.2–5.6)	6.6 (5.7–7.4)
Serum creatinine (mg/dL)	0.8 (0.7–0.9)	0.7 (0.6–0.8)	0.9 (0.9–1.0)
Total cholesterol (mg/dL)	181 (162–206)	180 (162–203)	184 (163–209)
Triglycerides (mg/dL)	142 (101–197)	132 (96–181)	158 (114–234)
Serum sodium (mmol/L)	141 (140–142)	140 (140–141)	141 (140–142)
**Nutritional variables** ^*a*^			
Total fructose (g/day)	50.9 (35.8–69.6)	50.1 (36.7–68.6)	52.3 (35.0–70.2)
Natural fructose (g/day)	20.4 (13.6–28.7)	21.7 (14.6–29.7)	18.8 (11.4–26.3)
Added fructose (g/day)	26.0 (17.0–42.7)	25.2 (15.4–39.9)	27.4 (17.0–46.0)
Sucrose (g/day)	34.8 (25.9–46.6)	33.8 (25.9–46.6)	36.0 (25.9–47.1)
Carbohydrates (g/day)	276 (223–336)	264 (215–328)	297 (241–349)
Total protein (g/day)	77 (61.9–92.9)	73.5 (60.8–88.7)	82.2 (66.0–100.9)
Animal protein (g/day)	41.4 (30.9–53.3)	39.7 (30.0–50.4)	46.5 (33.7–57.5)
Total fat (g/day)	89.2 (70.1–111.4)	84.8 (67.0–104.3)	96.7 (76.5–119.0)
Saturated fat (g/day)	25.2 (19.4–31.9)	23.6 (18.8–30.0)	27.4 (20.8–34.8)
Monounsaturated fat (g/day)	35.6 (27.2–45.4)	32.8 (26.4–42.9)	39.1 (30.3–49.3)
Polyunsaturated fat (g/day)	17.6 (14.1–24.0)	17.3 (13.7–24.0)	18.2 (14.5–24.0)
Fiber (g/day)	6.0 (4.8–7.5)	5.9 (4.8–7.5)	6.1 (4.6–7.6)
Total energy intake (kcal/day)	2130 (1741–2563)	2058 (1680–2483)	2302 (1859–2727)

^*a*^ Values are expressed as median (IQR, 25th–75th percentile). ^*b*^ Values are expressed as n (%). BMI, body mass index; WC, waist circumference; SBP, systolic blood pressure; DBP, diastolic blood pressure.

**Table 2 nutrients-17-03623-t002:** Comparison of energy intake, fructose sources, anthropometric, biochemical, and lifestyle variables across the three identified clusters. Values are expressed as means. Analysis of variance (ANOVA) *F* and *p*-values are reported.

Variable	Cluster 0	Cluster 1	Cluster 2	F	*p*-Value
**Energy intake**					
Total energy (kcal/day)	1660.4	2404.3	3543.7	1526.3	<0.001
Protein (kcal/day)	239.2	347.0	524.5	1230.8	<0.001
Carbohydrates (kcal/day)	874.9	1231.6	1715.5	613.8	<0.001
Fat (kcal/day)	546.4	825.7	1303.8	1104.4	<0.001
**Fructose intake**					
Natural fructose (g/day)	17.1	23.7	34.6	15.09	<0.001
Added fructose (g/day)	24.2	34.5	45.4	68.9	<0.001
Natural fructose (g/1000 kcal)	10.5	10.0	10.3	0.14	0.871
Added fructose (g/1000 kcal)	14.2	14.3	12.9	1.5	0.222
**Fructose sources**					
Natural juice (g/day)	1.8	2.3	9.4	5.2	0.006
Fruits (g/day)	15.3	21.4	25.3	14.6	<0.001
Cereals (g/day)	1.02	1.2	1.6	28.0	<0.001
Candies (g/day)	6.8	7.6	8.9	2.5	0.08
Soda (g/day)	19.7	22.8	28.9	9.7	<0.001
Natural juice (g/1000 kcal)	1.2	1.0	3.0	3.3	0.039
Fruits (g/1000 kcal)	9.3	9.0	7.3	1.7	0.183
Cereals (g/1000 kcal)	0.7	0.5	0.5	15.8	<0.001
Candies (g/1000 kcal)	4.2	3.2	2.6	6.5	0.002
Soda (g/1000 kcal)	12.2	9.5	8.3	7.6	0.001
**Anthropometry**					
Height (m)	1.61	1.63	1.64	10.0	<0.001
Waist circumference (cm)	94.3	95.4	98.5	6.5	0.002
Weight (kg)	75.2	77.6	81.3	9.6	<0.001
BMI (kg/$m^{2}$)	29.0	29.2	30.1	3.2	0.042
Age (years)	38.2	38.4	36.2	3.4	0.036
**Biochemical variables**					
Uric acid (mg/dL)	5.4	5.7	5.8	5.7	0.003
Serum creatinine (mg/dL)	0.7	0.8	0.8	8.1	<0.001
Glucose (mg/dL)	96.2	97.9	99.1	1.18	0.308
Total cholesterol (mg/dL)	185.1	184.5	183.9	0.1	0.930
Triglycerides (mg/dL)	160.6	173.4	170.4	1.46	0.233
Serum sodium (mmol/L)	140.7	140.4	140.7	0.95	0.386
Atherogenic index	2.8	2.9	2.9	0.8	0.456
**Sleep and lifestyle**					
Sleep alterations	26.78	27.15	27.29	0.05	0.952
Snoring	39.3	43.5	48.1	4.8	0.009
Sleep breathing risk	13.06	12.32	14.85	0.74	0.479
Sleep adequacy	52.36	53.50	54.93	0.47	0.625
Drowsy	28.20	28.59	30.93	1.46	0.232
Overall sleep quality	6.50	6.59	6.56	0.49	0.611
Sleep optimal	0.43	0.41	0.43	0.22	0.800
Smoked	0.61	0.67	0.79	7.39	0.001
Currently smoking	0.31	0.32	0.41	2.55	0.079
Daily smoker	0.11	0.10	0.16	1.93	0.145
Ex smoker	0.19	0.22	0.21	0.74	0.477
Passive smoker	0.39	0.39	0.45	0.85	0.429
Alcohol consumption	0.74	0.73	0.76	0.29	0.749
Energy drink consumption	0.07	0.08	0.12	1.78	0.169

**Table 3 nutrients-17-03623-t003:** Feature importance derived from XGBoost, HistGradientBoosting, and Random Forest. The SHAP column represents mean absolute Shapley values, while importance scores in the other models reflect impurity-based feature relevance.

XGBoost		HistGradientBoosting		Random Forest	
**Feature**	**SHAP**	**Feature**	**Importance**	**Feature**	**Importance**
BMI	1.0564	BMI	0.1506	BMI	0.1303
Alcohol	0.2563	WC	0.0640	NatJuicesFruct	0.0251
NatJuicesFruct	0.2314	Weight	0.0485	Alcohol	0.0239
TRIG	0.2182	NatJuicesFruct	0.0444	FatAnimal	0.0072
CandiesFruct	0.1929	TRIG	0.0325	CHOL	0.0063
CREA	0.1862	CandiesFruct	0.0285	Snoring	0.0063
SodaFruct	0.1587	SodaFruct	0.0274	GLU	0.0049
SleepBR	0.1584	IAT	0.0274	SleepQual	0.0034
CerealsFruct	0.1509	CerealsFruct	0.0260	HighAnxState	0.0034

SHAP, SHapley Additive exPlanations; BMI, Body Mass Index (kg/m^2^); Alcohol, Alcohol consumption; WC, Waist circumference (cm); NatJuicesFruct, Fructose from natural juices (g/day); TRIG, Triglycerides (mg/dL); CREA, Creatinine (mg/dL); CandiesFruct, Fructose from candies (g/day); SodaFruct, Fructose from soda (g/day); SleepBR, Sleep breathing problems; FatAnimal, Animal fat (g/day); CHOL, Total cholesterol (mg/dL); Snoring, Snoring frequency; GLU, Glucose (mg/dL); SleepQual, Sleep quality; HighAnxState, High anxiety (State); IAT, Atherogenic Index.

**Table 4 nutrients-17-03623-t004:** Comparison of model classification performance. Values are point estimates with 95% CIs (10-fold CV for XGBoost; bootstrap for Random Forest, HistGradientBoosting, and Logistic Regression).

Algorithm	Balanced Accuracy (95% CI)	Sensitivity (95% CI)	Specificity (95% CI)	F1 (95% CI)	AUC (95% CI)
XGBoost	81.8 (79.9–83.6)	88.1 (84.1–92.1)	75.4 (72.4–78.5)	82.8 (80.9–84.7)	0.893 (0.882–0.904)
Random Forest	87.51 (77.11–95.85)	96.92 (95.09–98.49)	78.09 (57.14–94.45)	97.83 (96.70–98.80)	0.979 (0.960–0.994)
HistGradient Boosting	88.41 (78.04–96.91)	99.09 (97.88–100.00)	77.74 (57.14–94.74)	98.94 (98.13–99.69)	0.992 (0.980–0.999)
Logistic Regression	77.17 (75.35–79.26)	73.33 (70.40–76.11)	81.00 (78.62–83.66)	76.09 (73.81–78.44)	0.772 (0.753–0.793)

**Table 5 nutrients-17-03623-t005:** Top15 logistic regression features associated with fructose intake and their statistical contribution. Arrows indicate the direction of association: ↑ positive, ↓ negative. Significant features were defined as p<0.05.

Feature	Coefficient	Direction	*p*-Value	Significant
Natural fructose (g/day)	−393.98	↓	<0.001	Yes
Fructose from fruits (g/day)	290.09	↑	<0.001	Yes
Fructose from natural juices (g/day)	259.46	↑	<0.001	Yes
Fructose from soda (g/day)	22.39	↑	0.001	Yes
Added fructose (g/day)	−20.15	↓	0.002	Yes
Fructose from candies (g/day)	10.61	↑	0.015	Yes
Fructose from cereals (g/day)	1.20	↑	0.083	No
Total carbohydrate (g/day)	−0.91	↓	0.107	No
Monounsaturated fat (g/day)	−0.70	↓	0.129	No
Vegetable fat (g/day)	0.65	↑	0.136	No
Alcohol consumption	0.53	↑	0.148	No
Total protein (g/day)	0.51	↑	0.152	No
Height ($m^{2}$)	−0.49	↓	0.165	No
Body mass index (kg/$m^{2}$)	0.40	↑	0.172	No
Currently smokes	0.37	↑	0.181	No

## Data Availability

The original contributions presented in the study are included in the article/Appendix A; further inquiries can be directed to the corresponding authors.

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
