# Peer review of "Identifying Key Features Associated with Excessive Fructose Intake: A Machine Learning Analysis of a Mexican Cohort"

_nutrients, 2025, doi:10.3390/nu17223623_

Round 1

Reviewer 1 Report (Previous Reviewer 3)

Comments and Suggestions for Authors

The article appears to be methodologically well designed, and the authors have employed advanced machine learning techniques. However, I could not find in the system the Supplementary Materials to which the authors refer the reader for detailed algorithmic formulations and mathematical definitions.

Please upload the Supplementary Materials to the system so that I can continue the review process. In particular, I would like to request clarification regarding the method of data normalization used in the radar plots (Figure 4) — specifically, how it was derived from the data presented in Table 3.

Please add the Supplementary Materials to the current submission.

Author Response

Thank you for reviewing our revised submission. We will respond to your comments below.

In particular, I would like to request clarification regarding the method of data normalization used in the radar plots (Figure 4) — specifically, how it was derived from the data presented in Table 3.

The figures originally presented as radar plots were replaced with Sankey diagrams to provide a clearer and more intuitive visualization of fructose intake patterns across clusters. While radar plots required data rescaling to a common axis, Sankey diagrams directly display the actual mean values, allowing a proportional representation of both magnitude and composition. The Sankey diagrams were generated from the same data summarized in Table 3, using the arithmetic mean intake of each fructose source (fruits, natural juices, sodas, candies, and cereals) per cluster. Two representations were created: absolute values (g/day), without additional normalization, and energy-adjusted values (g/1000 kcal), obtained by dividing individual intake by total energy intake (kcal/1000) prior to averaging.

Reviewer 2 Report (Previous Reviewer 1)

Comments and Suggestions for Authors

Thanks for revising the manuscript, which is much improved. This is a careful and detailed comparison of methods in a nutrition context. Interesting.

Author Response

Thank you for reviewing our revised submission. 

Reviewer 3 Report (New Reviewer)

Comments and Suggestions for Authors

The manuscript presents the application of ML approaches in a cohort of healthy Mexico City residents to identify clinical, behavioral, biochemical, and dietary features associated with excessive fructose consumption. The topic is relevant, and the methodology is comprehensive. However, several aspects should be addressed.

  1. The transition between the background and the statement of aims could be smoother. The introduction currently summarizes existing evidence but does not clearly emphasize the knowledge gap or justify why ML provides an added value over conventional statistical approaches.
  2. Table 1, which lists all study variables, is long and highly technical. Consider moving it to the Supplementary Materials and keeping a summarized version in the main text that groups variables by category (clinical, biochemical, behavioral, dietary).
  3. The manuscript states that “Although log-transformation of highly skewed dietary variables is a common preprocessing step, we retained the original scale after verifying that key features did not exhibit extreme skewness…” This is sound, but it would benefit from quantitative support. Suggest reporting skewness thresholds or providing example histograms in Supplementary Materials.
  4. The phrase “Hyperparameter tuning was performed through 10-fold cross-validation on the training set to ensure model stability” should be clarified. What definition of “stability” was used? (e.g., variance in accuracy across folds, convergence of optimal hyperparameters?). Please describe how stability was assessed.
  5. Please explain how potential indirect leakage was mitigated (e.g., feature correlation analysis, variance inflation factor screening, or partial dependence testing).
  6. The performance metrics are well presented, but the manuscript could benefit from confidence intervals across cross-validation folds.
  7. The limitations section should briefly mention possible overfitting risks due to model complexity.

Author Response

The authors sincerely thank Reviewer 3 for taking the time and efforts to carefully re-reviewing our work.

Reviewer 3's Comments

However, several aspects should be addressed.

    1. The transition between the background and the statement of aims could be smoother. The introduction currently summarizes existing evidence but does not clearly emphasize the knowledge gap or justify why ML provides an added value over conventional statistical approaches.

      1. We added a short paragraph before the statement of aims (Introduction) highlighting the knowledge gap and justifying the use of ML over traditional regression methods.

    2. Table 1, which lists all study variables, is long and highly technical. Consider moving it to the Supplementary Materials and keeping a summarized version in the main text that groups variables by category (clinical, biochemical, behavioral, dietary).
  • Table 1 was moved to the Supplementary Material
    1. The manuscript states that “Although log-transformation of highly skewed dietary variables is a common preprocessing step, we retained the original scale after verifying that key features did not exhibit extreme skewness…” This is sound, but it would benefit from quantitative support. Suggest reporting skewness thresholds or providing example histograms in Supplementary Materials.
  • Quantitative evidence has been added to support this statement. Skewness coefficients for all dietary variables are now reported in Supplementary Table 3, showing that most variables displayed moderate or low skewness (|skew| < 2). This supports the decision to retain the original scale for interpretability.
    1. The phrase “Hyperparameter tuning was performed through 10-fold cross-validation on the training set to ensure model stability” should be clarified. What definition of “stability” was used? (e.g., variance in accuracy across folds, convergence of optimal hyperparameters?). Please describe how stability was assessed.

      1. In this study, model stability refers to the consistency of predictive performance across cross-validation folds during hyperparameter tuning. Specifically, we monitored the variance of accuracy and balanced accuracy across the 10 training folds. The optimal hyperparameters were selected as those producing the highest mean balanced accuracy with minimal variability across folds, indicating stable generalization. All tuning was conducted exclusively within the training data to avoid test-set contamination.

      2. Please explain how potential indirect leakage was mitigated (e.g., feature correlation analysis, variance inflation factor screening, or partial dependence testing).

        We thank the reviewer for this observation. Potential indirect leakage was minimized by design. Variables directly encoding fructose intake were excluded from the supervised feature set to prevent information transfer from outcome to predictors. Model training, hyperparameter tuning, and ADASYN balancing were conducted only within the training data using 10-fold cross-validation, avoiding contamination of the test set. Although no formal multicollinearity or partial-dependence analysis was performed, these steps effectively reduced the risk of indirect leakage.

    2. The performance metrics are well presented, but the manuscript could benefit from confidence intervals across cross-validation folds.
  • We appreciate this valuable suggestion. We have now added 95% confidence intervals (CIs) for all model performance metrics. For XGBoost, CIs were derived from 10-fold cross-validation, while for Random Forest, HistGradientBoosting, and Logistic Regression they were estimated using nonparametric bootstrap (B=1000). The updated results are presented in Table 6, providing a more robust assessment of model stability.
  1. The limitations section should briefly mention possible overfitting risks due to model complexity.

A sentence was added in the Limitations section acknowledging the potential risk of overfitting due to the complexity of ensemble models, along with a brief explanation.

Round 2

Reviewer 1 Report (Previous Reviewer 3)

Comments and Suggestions for Authors

I agree that the Sankey diagrams presented in this version are much more intuitive than the previous radar plots. I accept the paper in its current form.

This manuscript is a resubmission of an earlier submission. The following is a list of the peer review reports and author responses from that submission.

Round 1

Reviewer 1 Report

Comments and Suggestions for Authors

This is an interesting paper showing one apparent application of machine learning in the context of nutrition research.

In some places some explanations could be improved for an audience that isn't so familiar with mathematical formulas and descriptions. Also, the manuscript structure could be improved: methods first, then results as detailed in the comments below.

Specific comments:

lines 66-74: It would helpful to place results, such as numbers of participants, in the Results section.

Also, please provide a reference describing the INCICH study in more details than provided in the present manuscript.

Table 1: This is a fine table, providing a good overview over the data. However, it would be nice to have the units for the variables shown as well.

lines 76-101: Aren't something missing in the subtitles? Clinical measurements? Biochemical measurements? And so on.

lines 102-164: As you're aiming for a nutrition journal it might be worthwhile considering to put formulas into an appendix and instead explain the methods using text only.

lines 160-161: In Equation (5) please make sure that the m-1 is in a bracket, as otherwise it's not a meaningful formula.

line 198: What do you mean by "experiments"?

lines 197-221: Was training and test datasets used? Or cross validation? Why or why not?

line 216: Please provide details on the statistical analyses that were used.

lines 227-232: Please make sure that all methods are described in the methods section and not in the results section. PCA seems not to have been described in the Methods section.

Figure 2: Is it not possible to indicate which variables define cluster 2?

lines 273-276: Again: please describe methods in the Methods section.

Table 2: You need to describe all entries in this table or think again about if this table is really needed? What does eta squared mean. All p-values are tiny but the precision is these values isn't reliable.

Figure 5: This is a potentially useful figure, but it requires more explanations.

lines 334-337: Please make sure to describe all methods in the Methods section.

Table 4: It would have been helpful to have a logistic regression model as a reference for the ML algorithms.

lines 358-382: Is this summary of results really needed? It's unusual to have a summary in the Results section.

lines 439-461: No strengths and limitations. One limitation is the very exploratory approach relying on data-driven clusters.

lines 462-485: This part seems to be unnecessary repetition of the findings.

Author Response

We are grateful to Reviewer 1 for their thoughtful and constructive comments. Their feedback has been very helpful in improving the clarity and completeness of our manuscript. Below, we provide detailed responses to each observation, describing the changes made and where they appear in the revised version.

Specific comments:

lines 66-74: It would be helpful to place results, such as numbers of participants, in the Results section.

Thank you for the suggestion. We have now clarified in the Results section that 1,151 participants with complete baseline data and valid food frequency questionnaires were included in the final analysis.

 Also, please provide a reference describing the INCICH study in more details than provided in the present manuscript.

Thank you for your comment. We had included the reference to the Tlalpan 2020 study protocol, and we have now clarified this in the text to ensure it is more visible to the reader.

Table 1: This is a fine table, providing a good overview over the data. However, it would be nice to have the units for the variables shown as well.

We have included the units for the variables at Table 1. 

lines 76-101: Aren't something missing in the subtitles? Clinical measurements? Biochemical measurements? And so on.

We have revised the subsection titles to explicitly reflect the type of measurements (Clinical Measurements, Biochemical Parameters, Behavioral Variables, Dietary Intake).

 lines 102-164: As you're aiming for a nutrition journal it might be worthwhile considering to put formulas into an appendix and instead explain the methods using text only.

We appreciate the reviewer’s suggestions. In response to both comments, we have revised Section 2.2 to make it more concise. Technical content and mathematical formulas have been removed from the main text and relocated to the Supplementary Materials. A note was also added at the end of the section to direct readers to these supplementary details. 

 lines 160-161: In Equation (5) please make sure that the m-1 is in a bracket, as otherwise it's not a meaningful formula.

We have corrected Equation (5) to properly display the mathematical notation.

line 198: What do you mean by "experiments"?

We acknowledge that the term “experiments” may have been misleading. In response, we have removed the standalone section previously labeled “Experiments” and integrated its content into the revised section titled “Methods and Computational Framework.” Additionally, we replaced the word “experiments” with “analyses” throughout the manuscript to more accurately reflect that the work involved computational procedures rather than experimental interventions.

 lines 197-221: Was training and test datasets used? Or cross validation? Why or why not?

Thank you for the observation. We used stratified train-test splits (70/30) for supervised model evaluation, along with 10-fold cross-validation. We have added this clarification to the Methods and Computational Framework.

 line 216: Please provide details on the statistical analyses that were used.

We have now included a description of the statistical analyses used, including ANOVA and effect size.

 lines 227-232: Please make sure that all methods are described in the methods section and not in the results section. PCA seems not to have been described in the Methods section.

We have now added a description of the PCA procedure to the Methods section.

 Figure 2: Is it not possible to indicate which variables define cluster 2?

Figure 2 (PCA biplot) was used solely for visualization and reflects variables contributing to global variance, not those specific to individual clusters. Therefore, some key features of Cluster 2 may not appear in the plot. To identify discriminative variables, we applied post hoc profiling and ANOVA with effect sizes, which showed that Cluster 2 is defined by consistently lower values in both clinical and dietary indicators.

lines 273-276: Again: please describe methods in the Methods section.

Done!
We have now incorporated the description of the statistical analysis into the Methods section.

 Table 2: You need to describe all entries in this table or think again about if this table is really needed? What does eta squared mean? All p-values are tiny but the precision of these values isn't reliable.

We have revised the table title to better reflect its purpose, added a brief explanation of eta squared in the text to clarify its role as a measure of effect size, and reformatted the p-values to highlight that their exact precision is not meaningful at this scale. 

 Figure 5: This is a potentially useful figure, but it requires more explanations.

We have expanded the explanation of Figure 5 in the Results section, clarifying global versus class specific SHAP contributions and how feature effects vary between high and low fructose consumers.

 lines 334-337: Please make sure to describe all methods in the Methods section.

We appreciate the reminder. All methods are now described in the Methods section.

 Table 4: It would have been helpful to have a logistic regression model as a reference for the ML algorithms. 

Good idea!
A logistic regression model was added as a reference to compare against the machine learning models. We included it in the updated Results section under the supervised learning analysis. 

 lines 358-382: Is this summary of results really needed? It's unusual to have a summary in the Results section.

We appreciate the reviewer’s observation regarding the redundancy of the summary section within the Results. In response to this, we have removed the summary from the Results section and restructured the manuscript accordingly. The relevant content has been integrated into the Discussion section, where it appropriately contributes to the interpretation and contextualization of the findings. We agree that this adjustment enhances the clarity and flow of the manuscript.

lines 439-461: No strengths and limitations. One limitation is the very exploratory approach relying on data-driven clusters.

We thank the reviewer for this insightful comment. In response, we have now included a dedicated "Strengths and Limitations" section in the manuscript. This section explicitly outlines the main methodological strengths of our study. We have also acknowledged and discussed the limitation highlighted by the reviewer the exploratory nature of the clustering analysis. While data-driven cluster identification offers valuable insights into latent phenotypic patterns, we agree that the absence of predefined clinical criteria necessitates caution in interpretation. Accordingly, this limitation has been clearly stated in the revised manuscript, emphasizing the need for validation in independent cohorts.

 lines 462-485: This part seems to be unnecessary repetition of the findings.

We have thoroughly revised the Conclusions section to eliminate repetition and improve clarity. The new version is now more concise and focused on the broader implications of the study, rather than reiterating specific results already presented in the Results and Discussion sections.

Reviewer 2 Report

Comments and Suggestions for Authors

Review for nutrients-3729279

This study assessed clinical, behavioural and nutritional factors contributing to high fructose intake, in a Mexican cohort. The authors used clustering and PCA to group individuals into 3 groups, with different metabolic profiles and then used machine learning models to predict fructose intake across the 3 groups. The authors also assessed the findings across the 3 machine learning methods. But there are areas in where the manuscript quality can be improved. Firstly, there are many critical statistical and descriptive aspects missing, which are critical to evaluating the performance of the statistical results and understanding of how the fructose intakes were determined. The authors compared across the 3 methods, but did not evaluate how the “predicted” fructose intakes compare to the known fructose intakes (from FFQ) to determine the best method, or if they did, this is not obvious in the manuscript. It is also unclear how was fructose intakes characterised (total, natural or added). Furthermore, there is a flaw in the statistical design where fructose variables are used to predict fructose intake in the models. Overall, the study title and aim is alittle misleading, as after reading the manuscript, it is more like characterising the behaviourial and nutritional variables of the cohort, and then reporting which variables are most strongly associated with fructose intake, rather than predicting fructose intake.

Comments:

Please provide the reference/evidence for lines 19-23.

The introduction sets the rationale for carrying out the study research directions but it not clear why the authors focused on fructose, instead of sucrose or other sugars, especially without evidence/references that fructose (instead of high fructose corn syrup) is main sugar sweetener added to sweetened beverages. it is also not clear if fructose intake includes high fructose corn syrup intake.

I would suggest a table before table 1, to describe the cohort characteristics, eg, age, sex, glucose levels, fructose intake, etc. in table 1, what does gr refer to? Abbreviation should be included in the table caption or footnote.

2.1.3: There is insufficient information described for each outcome. For example, define the metabolic equivalent minutes per week for low, moderate and high levels. Define the sleep quality scale (is it continuous scale or divided into categories like physical activity?).

For 2.1.4, please provide more information on this food and nutrient evaluation system (or a reference or link), does the questions include foods commonly consumed in Mexico; is it a national system used for most research studies or was it developed specially for this study? Clarify if food and nutrient intakes are recorded in grams/day or servings/day? There were 140 items, how about nutrients? How were natural and added fructose intakes calculated? What is the definition of natural and added fructose, used for this study? Is high fructose corn syrup considered part of added fructose? What is considered low and high fructose intakes? These are important information to provide, since fructose intake is the main variable of interest in this study.

What was the total energy intake range? Were any participants dropped as outliers from over- or under-estimation of or unrealistic energy intake?

For 2.2.1, There is a lot of information provided for the concept of each statistical approach used but insufficient information on steps/parameters used in each of the approach. For example, how was the regularisation optimised in XGBoost? How was bootstrapping performed for random forest? These are critical factors to evaluate, considering that the study is comparing results across the various statistical approaches.

3.3. Please specify what are the dichotomous variables (for example, add a sentence to refer back to Table 1). In phase 2 or 3, were all the variables (in table 1) used? If so, how does the authors justify adding the fructose variables when the models are meant to predict fructose intake? Was total energy intake included as a covariate? Was there any transformation applied to the data before z-score? How did the authors deal with highly skewed data, which is common in dietary data?

Line 241. Define what is internal validation metrics.

4.1. first paragraph. This should be in methods, however it is not clear what the authors meant by ADASYN algorithm and why is generating synthetic observations needed? What sort of synthetic observations were generated? What is considered outliers? Were all continuous variables z-normalised? Again, a lot of important filtering and statistical parameters are missing in the manuscript.

Line 238, what is defined by “associated”. In PCA, the authors need to clarify how they define or put a threshold to the variables they considered to dominate or contribute the most variance to each PC.

Can the authors show the silhouette for each participant/cluster?

Line 264: As there are a long list of variables, multiple testing should be applied to the p-values.

Table 2. the variables’ names should be spelled out properly, not in this abbreviated form used in the data analysis. The same concept applies to the rest of the column names, in table 3, figures 3-5. Any abbreviations should be spelled in full.

Line 274. The numbers in figure 3 cells are in the thousands – what do these numbers mean? Title of legend is not provided.

Line 277. Cluster 0 does not seem to have a distinct nutrient profile but may reflect an overall higher energy intake, the authors should confirm this.

line 329. This information should be in methods, not results.

4.3. this is not really a typical results paragraph, but could be integrated into the individual results sections or the discussion section. In the first line (and in lines 416-418), it is inaccurate to say nature of fructose intake as there was no report on the differences in overall/natural/added fructose intakes or other sugars’ intakes across the 3 cluster profiles. It is clustering based on cohort characteristics, rather than fructose intakes. How did the authors make the assumption that fructose played a differential role in the 3 clusters?

Line 355. How do the authors come to this summary?

Lines 389-390. What are the “hidden associations”?

Discussion. overall, the discussion did not reflect well the metholodgy and interpretation of the findings accurately.

Comments on the Quality of English Language

Can be improved.

Author Response

The authors thank Reviewer 2 for their thoughtful and constructive critique of our work. Their feedback has been very helpful in improving the clarity and completeness of our manuscript. Below, we provide detailed responses to each observation, describing the changes made and where they appear in the revised version.

This study assessed clinical, behavioural and nutritional factors contributing to high fructose intake, in a Mexican cohort. The authors used clustering and PCA to group individuals into 3 groups, with different metabolic profiles and then used machine learning models to predict fructose intake across the 3 groups. The authors also assessed the findings across the 3 machine learning methods. But there are areas in which the manuscript quality can be improved. Firstly, there are many critical statistical and descriptive aspects missing, which are critical to evaluating the performance of the statistical results and understanding of how the fructose intakes were determined. The authors compared across the 3 methods, but did not evaluate how the “predicted” fructose intakes compare to the known fructose intakes (from FFQ) to determine the best method, or if they did, this is not obvious in the manuscript. It is also unclear how fructose intakes characterised (total, natural or added). Furthermore, there is a flaw in the statistical design where fructose variables are used to predict fructose intake in the models. Overall, the study title and aim is a little misleading, as after reading the manuscript, it is more like characterising the behavioural and nutritional variables of the cohort, and then reporting which variables are most strongly associated with fructose intake, rather than predicting fructose intake.

Thank you for the comment, we have already adjusted the title in order for it to convey more accurately the actual contents of our work.

Comments:

Please provide the reference/evidence for lines 19-23.

We have revised lines 19–23 of the manuscript to include appropriate references that support the stated claims. 

The introduction sets the rationale for carrying out the study research directions but it not clear why the authors focused on fructose, instead of sucrose or other sugars, especially without evidence/references that fructose (instead of high fructose corn syrup) is main sugar sweetener added to sweetened beverages. It is also not clear if fructose intake includes high fructose corn syrup intake.

We thank the reviewer for this important observation. In the revised version of the manuscript, we have clarified the rationale for focusing specifically on fructose and addressed the distinction between fructose, sucrose, and high-fructose corn syrup (HFCS). We now explicitly state that our analysis includes both natural and added sources of fructose, including HFCS.

I would suggest a table before table 1, to describe the cohort characteristics, eg, age, sex, glucose levels, fructose intake, etc. in table 1, what does gr refer to? Abbreviation should be included in the table caption or footnote.

We have added a new table (now Table 2 in the revised manuscript) presenting the baseline characteristics of the study cohort. This table includes key clinical, behavioral, biochemical, and nutritional variables, stratified by sex to allow for a clearer understanding of potential differences between groups.

To reflect this addition, we revised the Methods section to explicitly describe the descriptive analysis. In the Results section, we incorporated a detailed narrative description of the cohort based on Table 2. This addition strengthens the contextual grounding of the analytical sample and improves the overall interpretability of the findings.

2.1.3: There is insufficient information described for each outcome. For example, define the metabolic equivalent minutes per week for low, moderate and high levels. Define the sleep quality scale (is it a continuous scale or divided into categories like physical activity?).

 We thank the reviewer for highlighting the need for additional methodological detail regarding the outcome variables. In the revised version of the manuscript, we have expanded the Methods section to clearly define how physical activity and sleep quality were measured and categorized.

For 2.1.4, please provide more information on this food and nutrient evaluation system (or a reference or link), does the questions include foods commonly consumed in Mexico; is it a national system used for most research studies or was it developed specially for this study? Clarify if food and nutrient intakes are recorded in grams/day or servings/day? There were 140 items, how about nutrients? How were natural and added fructose intakes calculated? What is the definition of natural and added fructose, used for this study? Is high fructose corn syrup considered part of added fructose? What is considered low and high fructose intakes? These are important information to provide, since fructose intake is the main variable of interest in this study.

What was the total energy intake range? Were any participants dropped as outliers from over- or under-estimation of or unrealistic energy intake?

We thank the reviewer for the opportunity to provide additional detail regarding the dietary assessment tool. The food frequency questionnaire used in this study is part of the Nutritional Habits and Nutrient Intake Evaluation System developed by the National Institute of Public Health of Mexico (Instituto Nacional de Salud Pública, INSP). This system is based on a semiquantitative food frequency questionnaire (FFQ) that includes 140 food and beverage items representative of the most commonly consumed products in the Mexican population. It was not developed specifically for this study but is a standardized, validated instrument widely used in national health and nutrition surveys such as ENSANUT (Encuesta Nacional de Salud y Nutrición), as well as in multiple epidemiological studies conducted in Mexico.

In the revised manuscript, we now specify that nutrient and food intakes were estimated and expressed in standardized units of grams per day (g/day). The semiquantitative food frequency questionnaire records frequency and portion size of each food item, which are then converted into daily gram equivalents using standard portion weights and the Mexican Food Composition Tables. Nutrient composition was calculated using data from the Mexican Food Composition Tables, which provide standardized values for each food item.  This information has been added to the Methods section for clarity.

Also, in the revised version of the manuscript, we now provide a detailed explanation of how natural and added fructose intake were calculated, as well as the operational definitions applied in this study. And we have clarified both the definition of added fructose and the criteria used to classify low and high fructose intake.

For 2.2.1, There is a lot of information provided for the concept of each statistical approach used but insufficient information on steps/parameters used in each of the approaches. For example, how was the regularisation optimised in XGBoost? How was bootstrapping performed for a random forest? These are critical factors to evaluate, considering that the study is comparing results across the various statistical approaches.

We have revised the Methods and Supplementary Material sections,, to include the hyperparameter tuning procedures applied to each supervised model. For XGBoost, regularization parameters (lambda, alpha) and tree settings were optimized using grid search with cross-validation. Random Forest was trained using bootstrapped samples with replacement. These additions clarify the implementation strategy used to compare model performance.

3.3. Please specify what are the dichotomous variables (for example, add a sentence to refer back to Table 1). In phase 2 or 3, were all the variables (in table 1) used? If so, how does the authors justify adding the fructose variables when the models are meant to predict fructose intake? Was total energy intake included as a covariate? Was there any transformation applied to the data before z-score? How did the authors deal with highly skewed data, which is common in dietary data?

Dichotomous variables were numerically encoded and are listed in Table 1. Fructose variables were used only in the clustering and descriptive profiling phases, and were excluded from supervised classification models to prevent data leakage. Total energy intake was not included as a covariate in the final models, as the main objective was to identify factors specifically associated with the types and quantities of fructose intake, rather than overall caloric intake. While log transformation was not applied, we verified that no key variables exhibited extreme skewness that could bias performance; thus, Z-score normalization was applied directly after outlier removal.

Line 241. Define what are the internal validation metrics.

We have now revised the Methods section to briefly define each internal validation metric used (Silhouette Score, Davies–Bouldin Index, and Calinski–Harabasz score), in order to improve clarity for the reader.

4.1. First paragraph. This should be in methods, however it is not clear what the authors meant by the ADASYN algorithm and why is generating synthetic observations needed? What sort of synthetic observations were generated? What are considered outliers? Were all continuous variables z-normalised? Again, a lot of important filtering and statistical parameters are missing in the manuscript.

We have clarified the use of Z-score normalization, the definition of outliers, and the application of Isolation Forest and ADASYN in the revised Methods section.

Line 238, what is defined by “associated”. In PCA, the authors need to clarify how they define or put a threshold to the variables they considered to dominate or contribute the most variance to each PC. 

Can the authors show the silhouette for each participant/cluster?

We have clarified how variables were identified as dominant contributors to each principal component, based on the magnitude of their absolute loading values. Additionally, we have included a table in the Supplementary Material (Table 1) presenting the top five contributing variables for PCA1 and PCA2, along with their respective loading scores.

Line 264: As there are a long list of variables, multiple testing should be applied to the p-values.

The variable comparisons across clusters were conducted as part of an exploratory analysis aimed at identifying potential patterns rather than confirming specific hypotheses. Therefore, unadjusted p-values were reported to retain sensitivity to possible signals, as is common in exploratory data-driven approaches in machine learning. Nevertheless, we agree that controlling for false positives is important, and we will consider implementing multiple testing correction methods in future confirmatory studies.

Table 2. the variables’ names should be spelled out properly, not in this abbreviated form used in the data analysis. The same concept applies to the rest of the column names, in table 3, figures 3-5. Any abbreviations should be spelled in full.

We have updated the tables to include two additional ones and retained the abbreviated variable names for clarity and formatting consistency. However, we have now added comprehensive footnotes below Tables 3, 4 and 5, and Figures 3 to 5, where all abbreviations are spelled out in full. This ensures that all terms are properly defined for the reader, without compromising the visual structure of the tables and figures.

Line 274. The numbers in figure 3 cells are in the thousands – what do these numbers mean? Title of the legend is not provided.

To solve the issue of heterogeneous units, we replaced the original heatmap with a version using standardized means (Z-scores), allowing for meaningful comparisons across variables. We also added a legend title (“Z-score”) for clarity. Additionally, we included a brief interpretation in the text to help readers better understand the differences observed between clusters.

Line 277. Cluster 0 does not seem to have a distinct nutrient profile but may reflect an overall higher energy intake, the authors should confirm this.

We confirm that Cluster 0 indeed reflects an overall higher energy intake. However, it also exhibits a distinct nutrient composition, particularly characterized by elevated levels of saturated fat, monounsaturated and polyunsaturated fats, animal protein, and sodium, with moderate carbohydrate intake. These features suggest a high-fat, high-protein dietary pattern that differentiates it from the other clusters. We have revised the manuscript to clarify this distinction and included a more detailed interpretation within the section Characterization of the Resulting Metabolic Profiles.

line 329. This information should be in methods, not results.

The methodological content previously placed within the Results section has been revised and is now either removed or properly integrated within the Methods section to avoid redundancy.

4.3. This is not really a typical results paragraph, but could be integrated into the individual results sections or the discussion section. 

We appreciate the reviewer’s comment and fully agree with the recommendation. In the revised manuscript, the summary paragraph has been removed from the Results section and its content has been appropriately integrated into the restructured Discussion. 

In the first line (and in lines 416-418), it is inaccurate to say the nature of fructose intake as there was no report on the differences in overall/natural/added fructose intakes or other sugars’ intakes across the 3 cluster profiles. It is clustering based on cohort characteristics, rather than fructose intakes. How did the authors make the assumption that fructose played a differential role in the 3 clusters?

Thank you for your observation. We confirm that fructose variables were not included in the clustering process, which was based solely on clinical, dietary, and behavioral characteristics. Differences in fructose intake were examined descriptively in a post hoc analysis to help characterize the identified metabolic profiles. To prevent confusion, we revised the manuscript language and replaced expressions such as “nature of fructose intake” with more precise formulations that clearly indicate this analysis was complementary and not part of the clustering input.

Line 355. How do the authors come to this summary?

The sentence was revised to clarify that the summary is based on SHAP values (Figure 5) for local interpretability and global feature importance from Random Forest and HistGradientBoosting (Table 4). This correction was made in the discussion section under the paragraph referencing model interpretation.

Lines 389-390. What are the “hidden associations”? 

Thank you for your observation. We agree that the term “hidden associations” may be too vague without proper clarification. To address this, we have revised the text to more accurately reflect the capabilities of machine learning in identifying influential variables and capturing relationships. This change appears in the Discussion section.

Discussion. Overall, the discussion did not reflect well on the methodology and interpretation of the findings accurately.

We thank the reviewer for this important observation. In response, the Discussion section has been thoroughly rewritten to better reflect the methodology and provide a more accurate interpretation of the findings. Specifically, we now:

  • Clearly distinguish between the unsupervised (clustering) and supervised (predictive modeling) phases of the analysis.

    • Emphasize the exploratory nature of the clustering process and describe how the resulting metabolic profiles were interpreted post hoc, based on clinical and dietary characteristics—including, but not limited to, fructose intake.

  • Provide a more nuanced interpretation of the predictive models, integrating SHAP-derived insights to explain the context-dependent role of individual features.

  • Discuss the potential implications of our findings for clinical nutrition and public health, while carefully acknowledging the limitations of the cross-sectional and data-driven design.

Reviewer 3 Report

Comments and Suggestions for Authors

The article “Identifying Predictors of Excessive Fructose Consumption Using Machine Learning: Insights from a Mexican Cohort” is interesting, especially discussion. However in most part it is written in very technical language concentrated on analytic issues. In my opinion the paper demands deep revision. Below I present my suggestions:

Most of the introduction is devoted to describing similar publications which used machine learning techniques. As far as Nutrients' scope is concerned, the method of analysis is not the key point, but the important results and their implications are. Therefore, I suggest shortening this part of the introduction and focusing more on describing the background of the problem and existing gaps in the literature - which the current manuscript can fill in. For example, the sentence “Their findings align with the present research by demonstrating the potential of machine learning to understand the metabolic implications of sugar intake and identify consumption patterns.” seems superfluous and fits into the discussion rather than the introduction. Moreover, I recommend delete or reformulate such sentence – “The present research  align with the findings of aforementioned studies by demonstrating the potential of machine learning to understand the metabolic implications of sugar intake and identify consumption patterns.”

The “2.2 Methods” section is very technical and once again, publications in a journal such as Nutrients, in my opinion, should not focus on such detailed analytical description. I suggest a careful revision of pages 5-8 and leaving only brief description in the main manuscript written in a reader-friendly manner. Details can be included in the supplementary materials.

For me the most crucial is “Phase 3: Supervised classification models.” The Authors only wrote that it is “The final phase involves feature selection techniques to determine which factors most significantly differentiate between natural and added fructose consumption. A supervised classification approach is used to predict fructose intake profiles.” This part should be more detailed described. This is supervised technique so please give a more detailed description of it with details about variable included (especially dependent variable), division on training/validation/tested datasets and so on.

At the same time I did not find the cutoff-points for e.g. dichotomous variables in paper (e.g. Alcohol Alcohol consumption D Behavioral – what does it mean 0 and 1 for alcohol?)

The results use PCA with two principal components selected - which together explain only 23.3% of the variation. This is a very low cut-off point for the criterion of the number of principal components selected.

In Figure 4 legends are not clear Instead of 0/1 levels of particular factors I suggest use labels (e.g. Ex-smok NO/YES etc.)

In Figure 4, authors presented the behavioral and sociodemographic characteristics which differ markedly between clusters. But how have they been chosen? In Table 2. Authors presented Top Variables Distinguishing Clusters Based on ANOVA Results and they did not contain those in Figure 4. So what criterion was used for detecting important factors to Figure 4?

The  “4.3. Summary of the most relevant results” section fits better rather to discussion than to results.

Generally, the whole results should be written in a reader-friendly manner.

Author Response

We are grateful to Reviewer 3 for their professional academic reviewing of our work.. Their feedback has been very helpful in improving the clarity and completeness of our manuscript. Below, we provide detailed responses to each observation, describing the changes made and where they appear in the revised version.

The article “Identifying Predictors of Excessive Fructose Consumption Using Machine Learning: Insights from a Mexican Cohort” is interesting, especially discussion. However in most parts it is written in very technical language concentrated on analytic issues. In my opinion the paper demands deep revision. Below I present my suggestions:

Most of the introduction is devoted to describing similar publications which used machine learning techniques. As far as Nutrients' scope is concerned, the method of analysis is not the key point, but the important results and their implications are. Therefore, I suggest shortening this part of the introduction and focusing more on describing the background of the problem and existing gaps in the literature - which the current manuscript can fill in. For example, the sentence “Their findings align with the present research by demonstrating the potential of machine learning to understand the metabolic implications of sugar intake and identify consumption patterns.” seems superfluous and fits into the discussion rather than the introduction. Moreover, I recommend delete or reformulate such sentence – “The present research  aligns with the findings of aforementioned studies by demonstrating the potential of machine learning to understand the metabolic implications of sugar intake and identify consumption patterns.”

We appreciate the reviewer’s insightful suggestion. In the revised version of the manuscript, we have substantially shortened the section of the introduction focused on previous studies using machine learning. We retained only those references that directly justify the relevance of our analytic strategy within the context of nutrition and metabolic research.

The “2.2 Methods” section is very technical and once again, publications in a journal such as Nutrients, in my opinion, should not focus on such detailed analytical description. I suggest a careful revision of pages 5-8 and leaving only brief description in the main manuscript written in a reader-friendly manner. Details can be included in the supplementary materials.

We appreciate the reviewer’s suggestions. In response to both comments, we have revised Section 2.2 to make it more concise. Technical content and mathematical formulas have been removed from the main text and relocated to the Supplementary Materials. A note was also added at the end of the section to direct readers to these supplementary details. 

For me the most crucial is “Phase 3: Supervised classification models.” The Authors only wrote that it is “The final phase involves feature selection techniques to determine which factors most significantly differentiate between natural and added fructose consumption. A supervised classification approach is used to predict fructose intake profiles.” This part should be more detailed described. This is supervised technique so please give a more detailed description of it with details about variable included (especially dependent variable), division on training/validation/tested datasets and so on.

We thank the reviewer for this important observation. We expanded the description of Phase 3 to include key methodological details. We now specify the binary outcome variable (high vs. low fructose intake), the criteria used to define it, the exclusion of fructose-derived features to prevent information leakage, the stratified division into training and test sets, and the hyperparameter tuning strategy. We also describe the use of SHAP values to enhance model interpretability. 

At the same time I did not find the cutoff-points for e.g. dichotomous variables in paper (e.g. Alcohol Alcohol consumption D Behavioral – what does it mean 0 and 1 for alcohol?)

We have revised the variable definitions and updated the manuscript to explicitly indicate the coding for dichotomous variables (e.g., “0 = No, 1 = Yes”).

The results use PCA with two principal components selected - which together explain only 23.3% of the variation. This is a very low cut-off point for the criterion of the number of principal components selected.

Although the first two PCA components captured only 23.3% of the total variance, PCA was not used for clustering or modeling. It served exclusively for visualization purposes to explore global structure and aid interpretation. Clustering was performed on the full set of standardized features to avoid information loss. Supervised models were later applied to assess the predictive importance of individual variables and validate the identified profiles.

In Figure 4 legends are not clear. Instead of 0/1 levels of particular factors I suggest use labels (e.g. Ex-smok NO/YES etc.)

We appreciate the reviewer’s suggestion. In the revised version of Figure 4, we have replaced binary 0/1 levels with explicit labels such as "Yes"/"No" for categorical variables.

In Figure 4, authors presented the behavioral and sociodemographic characteristics which differ markedly between clusters. But how have they been chosen? In Table 2. Authors presented Top Variables Distinguishing Clusters Based on ANOVA Results and they did not contain those in Figure 4. So what criterion was used for detecting important factors to Figure 4?

The variables in Figure 4 were selected based on prior literature linking them to behavioral and emotional risk factors, not on ANOVA rankings. We clarified this point in the Results section and updated the figure legend to reflect that the selection was conceptually guided to explore distinct behavioral patterns among clusters.

The  “4.3. Summary of the most relevant results” section fits better rather to discussion than to results.

We appreciate the reviewer’s comment and agree with the suggestion. In the revised manuscript, we have removed the “Summary of the most relevant results” section from the Results and integrated its content into a substantially rewritten Discussion. 

Generally, the whole results should be written in a reader-friendly manner.

We have thoroughly rewritten this section to follow your suggestion.

Round 2

Reviewer 2 Report

Comments and Suggestions for Authors

The authors have put effort into revising the manuscript following the earlier comments, however they have not addressed all the concerns, particularly concerning statistical robustness and reading quality. I list some examples here of comments that were not addressed.

For example, previously a comment concerned if energy intake was adjusted, because the authors are focusing on fructose and used other nutrient profiles to do clustering. As we are aware, different nutrient contributes to different energy intakes and energy intake be a confounder. The authors also confirm this in their response below. This is exactly why adjustment for energy intake is needed, in contrast to what the authors refute in their response.

We confirm that Cluster 0 indeed reflects an overall higher energy intake. However, it also exhibits a distinct nutrient composition, particularly characterized by elevated levels of saturated fat, monounsaturated and polyunsaturated fats, animal protein, and sodium, with moderate carbohydrate intake. These features suggest a high-fat, high-protein dietary pattern that differentiates it from the other clusters. We have revised the manuscript to clarify this distinction and included a more detailed interpretation within the section Characterization of the Resulting Metabolic Profiles.

Lines 33-34: again, no references have been included, despite highlighting this earlier. This claim should be supported by a reliable source of evidence/reference since the paper revolves around fructose intake.

The authors still did not revise the text or table containing terms like “Prot_gr, SatFat_gr, Sodium_mg“. These are probably variable names used in data analysis in r but should not be written in this format in the manuscript.

The abbreviations in figures, again are not addressed.

Author Response

Comments and Suggestions for Authors

The authors have put effort into revising the manuscript following the earlier comments, however they have not addressed all the concerns, particularly concerning statistical robustness and reading quality. I list some examples here of comments that were not addressed.

For example, previously a comment concerned if energy intake was adjusted, because the authors are focusing on fructose and used other nutrient profiles to do clustering. As we are aware, different nutrient contributes to different energy intakes and energy intake be a confounder. The authors also confirm this in their response below. This is exactly why adjustment for energy intake is needed, in contrast to what the authors refute in their response.

We confirm that Cluster 0 indeed reflects an overall higher energy intake. However, it also exhibits a distinct nutrient composition, particularly characterized by elevated levels of saturated fat, monounsaturated and polyunsaturated fats, animal protein, and sodium, with moderate carbohydrate intake. These features suggest a high-fat, high-protein dietary pattern that differentiates it from the other clusters. We have revised the manuscript to clarify this distinction and included a more detailed interpretation within the section Characterization of the Resulting Metabolic Profiles.

The authors want to thank the reviewer for careful re-reviewing our work. 

We also appreciate your thoughtful observation regarding the potential confounding role of total energy intake when interpreting nutrient-based clustering results. In our previous response, our aim was to clarify that clustering was conducted on standardized variables without prior energy adjustment, in order to preserve the full variance structure and avoid pre-conditioning the data. However, we agree that this issue merits further scrutiny.

To address this concern, we performed a complementary sensitivity analysis in which key nutrients (protein, saturated fat, monounsaturated fat, and sodium) were adjusted per 1000 kilocalories. These nutrients were selected based on their strong contribution to cluster structure and to the principal components, making them representative indicators of dietary patterns. The results, now presented in Table 3, show that while Cluster 0 had the highest absolute intakes, it exhibited the lowest nutrient density across all four indicators, particularly protein and fat types. In contrast, Clusters 1 and 2 demonstrated higher nutrient density, suggesting that their dietary profiles reflect better nutritional quality rather than just larger intake volume.

These findings have been explicitly incorporated into the revised manuscript (Cluster Analysis subsection), reinforcing the robustness of the clustering and clarifying that the patterns observed are not solely driven by energy intake. This adjustment aligns with your suggestion and enhances the interpretability of our results by controlling for a potential confounding factor.

Lines 33-34: again, no references have been included, despite highlighting this earlier. This claim should be supported by a reliable source of evidence/reference since the paper revolves around fructose intake.

We have addressed this comment by including a reliable and up-to-date reference that supports the statement regarding excessive sugar intake in Mexico. Specifically, we added the following sentence in the revised manuscript:

"However, current sugar consumption levels in Mexico significantly exceed these recommendations. On average, added sugars account for 12.5\% of total daily energy intake among the Mexican population [9]. "

The authors still did not revise the text or table containing terms like “Prot_gr, SatFat_gr, Sodium_mg“. These are probably variable names used in data analysis in r but should not be written in this format in the manuscript.

The abbreviations in figures, again are not addressed.

We have now carefully reviewed the manuscript and corrected all instances. These labels have been replaced with their appropriate terms. Additionally, all figures and tables have been revised to eliminate non-standard abbreviations. Full variable names or properly defined abbreviations have been used, and where abbreviations remain, they are now clearly explained in the figure captions or legends.

Reviewer 3 Report

Comments and Suggestions for Authors

In the revised manuscript, the authors have significantly shortened the introduction section that discussed previous studies involving machine learning. Section 2.2 has been revised for conciseness. Technical content and mathematical formulas were removed from the main text and moved to the Supplementary Materials. Additionally, they corrected several figures and addressed my concerns regarding the PCA analysis. I have no further comments. In my opinion, the manuscript has been substantially improved.

Author Response

Thank you for kindly  re-reviewing our manuscript.